# Electromagnetic Safety of Short-Range Radio Frequency Identification Systems

Slawomir Musial [1], Andrzej Firlej [1], Ireneusz Kubiak [1,*] and Tomasz Dalecki [2]

[1] Department of Electromagnetic Compatibility, Military Communication Institute—National Research Institute, 05-130 Zegrze Poludniowe, Poland; s.musial@wil.waw.pl (S.M.); a.firlej@wil.waw.pl (A.F.)

[2] C4I Systems Department, Military Communication Institute—National Research Institute, 05-130 Zegrze Poludniowe, Poland; t.dalecki@wil.waw.pl

* Correspondence: i.kubiak@wil.waw.pl

**Abstract:** The intensive development of information and telecommunications technology has a very large impact on society. On the one hand, it greatly facilitates many activities in everyday life, such as searching for information, establishing contacts with other people, or controlling household appliances at a distance. On the other hand, it poses serious threats to our personal data, the information we process, or our property. One of the examples of such threats may be radio identification systems, enabling the registration of working time, entry into restricted zones in workplaces and offices, or warehouse data. Copying data from an identification card may allow unauthorized persons access to premises and data that they should not have. The article presents the principles of operation, the results of the conducted research, and their analysis in the aspect of security of the short-range radio frequency identification systems used in relation to the RFID 125 kHz system, broadly used in access control systems or time and attendance systems. Particular attention has been paid to the possibility of unauthorized acquisition of information contained in the identification card in order to, for example, copy it and gain access to specific protected zones. An analysis of the security of such systems was carried out not only in relation to the data carriers themselves but also to complete access control systems installed in buildings. The research focused especially on the ability to determine the range of information penetration, i.e., the distance of remote information acquisition using electromagnetic radiated emissions.

**Keywords:** electromagnetic compatibility; electromagnetic disturbances; device immunity; measurement techniques; protection of information; NFC; RFID

## 1. Introduction

Time and attendance and access control systems are systems that increase the level of security and are used to identify people (goods) [1,2] moving around the facility and give them appropriate permissions [3,4]. With the help of the access control system (ACS), you can limit access to selected rooms or security zones, register and manage guest passes [5], and register vehicles moving in the protected area [6]. ACS includes two platforms: hardware and software. The hardware platform is primarily controllers, readers, barriers, gates, electromagnetic bolts, electromagnets, identifiers, etc. The software platform is actually a database that allows you to identify people (objects) with the permissions granted to them regarding staying in selected zones, and an executive application controls the hardware platform.

These ACS operate on the basis of wireless, near-range communication technology. The theoretical range (distance between the identifier and the reader) of such communication in such systems does not exceed several centimeters.

Radio frequency identification systems include:

- Radio Frequency Identification Low Frequency (RFID LF): access control and animal identification.

- Radio Frequency Identification High Frequency (RFID HF): access control, electronic tickets, tolls, micropayments [7,8], and loyalty programs.
- Radio Frequency Identification Ultra High Frequency (RFID UHF): product identification, warehouse management, logistics and transportation [9], supply chain control.
- RFID UHF: product identification, warehouse management, logistics and transportation, and supply chain control [10].
- RFID 2.4 GHz: identification and location of goods.

In the further part of the article, issues related to selected hardware solutions, the principles of operation of readers and identifiers, as well as the security of identifiers and ACSs based on RFID LF technology, are presented.

The above examples of RFID system applications show the possibilities of using the technology in various areas of life. The authors of the publication do not focus on data protection issues. Often, however, RFID systems process information that can determine the security not only of individuals but also of companies or the state. Hence, the authors of the article attempted to conduct comprehensive research and analysis of the obtained results in terms of the possibility of carrying out the process of electromagnetic infiltration of RFID systems. If the systems were found to be immune to electromagnetic eavesdropping, scenarios that could increase the effectiveness of RFID systems as sources of unwanted electromagnetic emissions were analyzed.

The aim of the study was to determine the possibility of increasing the range of radio communication to a value that allows unauthorized persons to read the data from the identifier. For this purpose, various electronic circuits were used to increase the level of the signal emitted by the identifier, to improve the signal quality by increasing the modulation depth factor, and to analyze the signals emitted by the complete access system when the identifier is read. In order to reduce undesirable electromagnetic disturbances that may come from the elements of the access system and may have an adverse effect on the measurement results, magnetic coils (receiving and transmitting) of our own design were used, cooperating with measuring equipment with low levels of internal electromagnetic disturbances. In addition, the measurements were carried out in a shielding anechoic chamber, which ensured separation from the external electromagnetic environment (Figure 1).

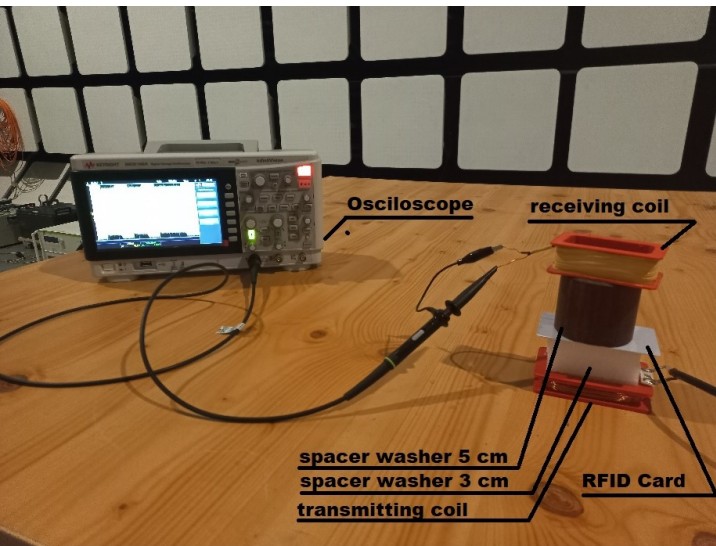

**Figure 1.** Actual measurement system.

## 2. General Information about RFID LF Systems

The basic ACS consists of:

- Controller;
- Card reader;

- Electromagnetic lock;
- A computer with the appropriate application and database;
- Identifiers in the form of cards, key rings, stickers, etc.

A simple diagram of such a system is shown in Figure 2.

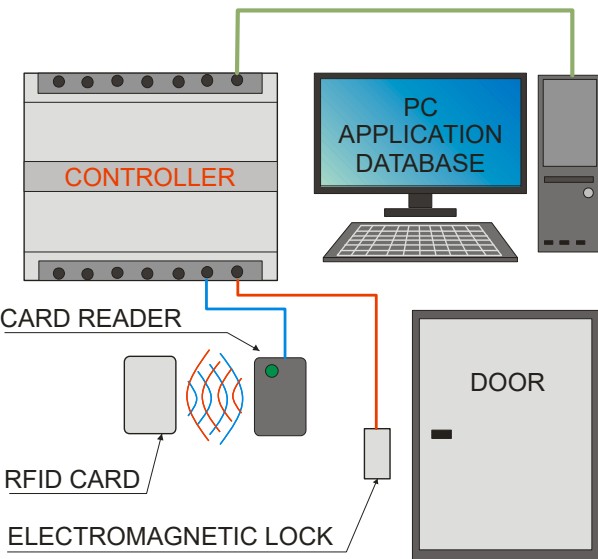

**Figure 2.** Diagram of the access control system.

A dedicated application for the controller and database is installed on the computer. It allows you to grant permissions for individual identifiers, determine the zone in which the identifier is (or was located), operate the database of individual identifiers with allowed zones, and enter data into the controller, which can then control access to the rooms in a limited way. In addition, the computer can also be used to program identifiers using special programmers (most often a reader with a recording function).

The most important function is performed by the controller. Based on the data programmed in it and data from the database, it controls actuators such as electromagnetic locks, barriers, or gates, allowing or not moving between different zones. The identity (ID) data is sent to the controller from the card reader (after touching the ID to it). Information about accepting or rejecting the entry request is sent to the application on the computer.

The card reader generates the magnetic field (in the case of the 125 kHz RFID system) necessary to power the card's electronics. After approaching the identifier, the received signal modulated by the identifier is demodulated and decoded in the reader, and then it is sent to the controller via the data bus. A card reader is usually made of one transmitting coil and one receiving coil. A reader with one transceiver is also a common solution. An example of a reader view is shown in Figure 3.

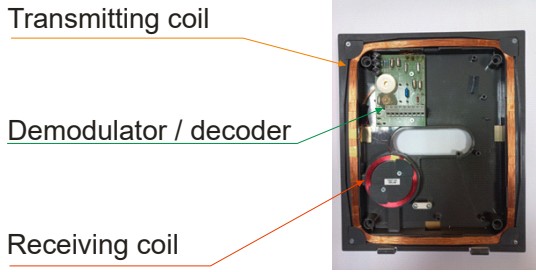

**Figure 3.** RFID LF card reader with one transmitting coil and one receiving coil.

The electromagnetic lock (bolt and gate) performs an executive function. After applying voltage from the controller, it unlocks the door, bolts, or allows the gate to rotate.

The lock itself usually does not send a feedback signal to the controller. Additional elements (if necessary), such as magnetic sensors, are used to determine whether the door has been opened.

The identifier is most commonly found in the form of a plastic card or key ring (Figure 4). It consists of a coil embedded in plastic and an electronic system (chip) [11], whose task is to encode the data stored in its memory and, on that basis, key the coil. The identifiers in the RFID LF system do not have their own power source. To power them [12,13], the magnetic field generated by the reader is used [14]. The most commonly used electronic circuit is the EM4100 family integrated circuit with 64-bit memory [15,16]. The memory structure of this chip is shown in Figure 5. Data from the system is output in series, continuously, as long as the identifier is in a magnetic field. Before sending, the data is encoded in the Manchester system [17], as shown in Figure 6.

|     |     |
| --- | --- |
| (**a**) | (**b**) |

**Figure 4.** RFID LF system identifier: (**a**) card and (**b**) keychain.

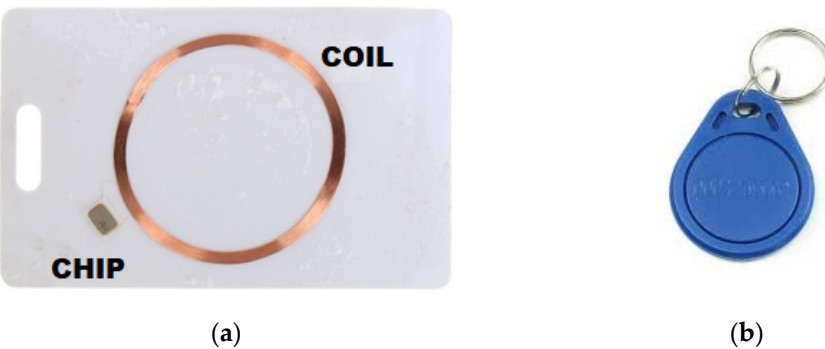

**Figure 5.** Structure of data stored in the identifier layout.

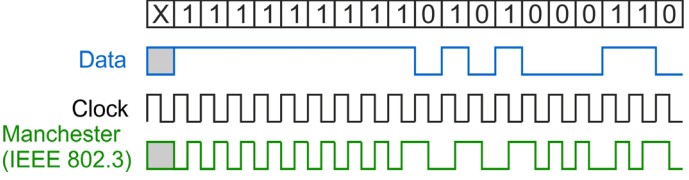

**Figure 6.** Manchester coding [18].

## 3. Modulation Depth Factor as a Function of Magnetic Field Strength

The modulation depth factor $m$ is one of the most important parameters of the $U$ signal determining its readability. It is defined by two signal parameters—$U_{max}$ and $U_{min}$ (Figure 7), and its value is calculated according to the relationship (1):

$$m = \frac{U_{max} - U_{min}}{U_{max} + U_{min}},$$

(1)

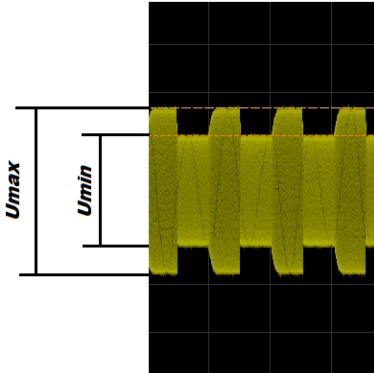

**Figure 7.** Parameters defining the modulation depth factor.

The value of the modulation depth factor is often expressed as a percentage (2):

$$m_\% = 100 \frac{U_{max} - U_{min}}{U_{max} + U_{min}},$$  (2)

The values of the modulation depth factor determined during the measurements are similar to those presented in [19] for the case of the nonlinear transponder model. In the case of modulation depth factor *m* with small values (for example, of the order of 2%), the correct reading of the *U* signal becomes difficult (Figure 8).

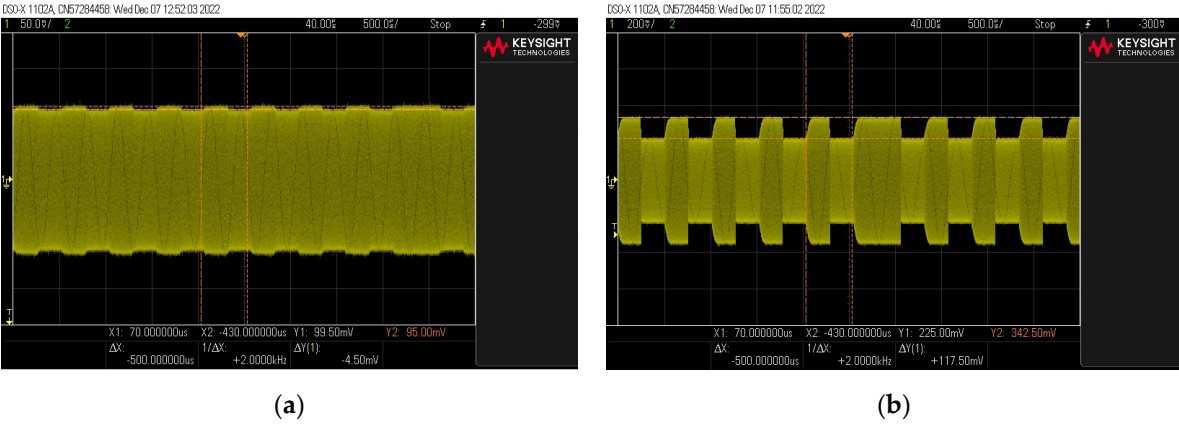

(**a**)                                        (**b**)

**Figure 8.** Example of a signal waveform with a modulation depth factor of (**a**) 2% and (**b**) 20%.

For the research aimed at determining the dependence of the value of the modulation depth factor *m* on the value of the magnetic field *H* acting on the RFID card, self-prepared coils—transmitting and receiving—were used [20–22]. It allowed us to eliminate additional unnecessary electronic circuits that could adversely affect the observed signals. A signal generator was connected to the transmitting coil, which during the measurements fed a *U* signal with adjustable amplitude and a nominal frequency of the card—125 kHz.

As noted in the introduction of the article, the tested RFID system, due to the value of the operating frequency and areas of application, may not be susceptible to electromagnetic infiltration. However, just like in the case of IT systems that process information in graphic form (laser printers, computers, laptops, and multifunctional devices), the RFID system can be subjected to external factors that change the operating parameters of electronic systems. In this way, additional sources of unwanted emissions are activated, or the effectiveness of existing ones is increased.

The main objective of the measurements was to check whether increasing the magnetic field (the strength of the signal delivered to the transmitting coil) acting on the identifier would increase the communication range. In this case, the communication range is the

same as a sufficiently large value of the modulation depth factor necessary for the correct detection of the received signal.

The receiving coil was connected to an oscilloscope with an impedance of 1 MΩ. After each measurement, the value of the modulation depth factor $m$ was determined using an oscilloscope, and then the receiving coil was replaced by a magnetic antenna connected to the terminals of the spectrum analyzer, and the value of the magnetic field strength $H$ corresponding to the previously determined modulation factor $m$ was read (taking into account the correction factors of the antenna). During these measurements, as shown in [23], a very high attenuation of the electromagnetic field was noticeable in the case of increasing the distance d and a high sensitivity to changes in the angle of the coils in relation to each other.

The measuring systems used for these tests are shown in Figures 9 and 10.

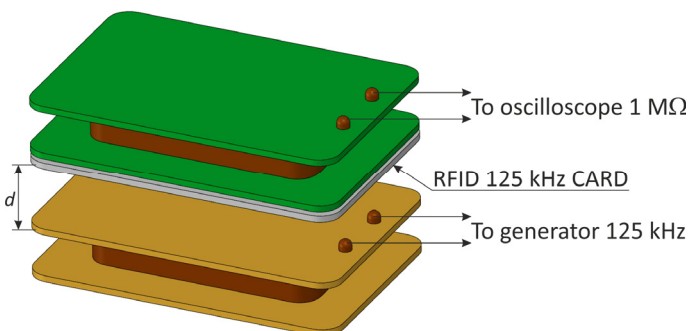

**Figure 9.** The measuring system for determining the modulation depth factor $m$.

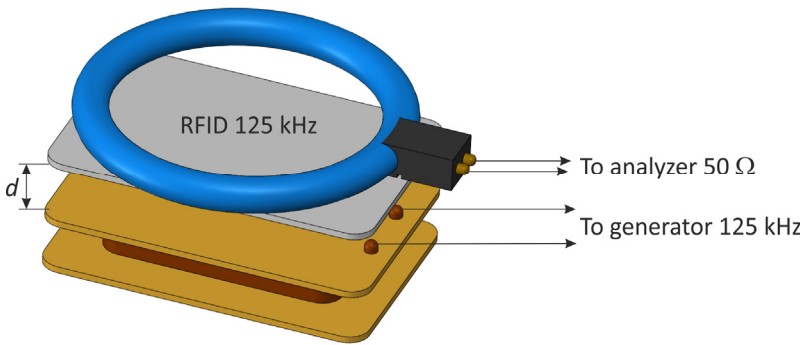

**Figure 10.** The measuring system for determining the generated magnetic field $H$.

The measuring apparatus listed in Table 1 was used for the tests.

**Table 1.** The apparatus used to determine the minimum magnetic field necessary to excite an RFID card.

| Equipment | Type | Producer |
| --- | --- | --- |
| Spectrum Analyzer | 3588A | Hewlett Packard (Palo Alto, CA, USA) |
| Spectrum Analyzer | FSL18 | Rohde and Schwarz (Munich, Germany) |
| Signal Generator | SNGL | Rohde and Schwarz (Munich, Germany) |
| Oscilloscope | DSOX1102A | Keysight (Santa Rosa, CA, USA) |
| Magnetic antenna | 94605-1 | Ailtech (Santa Rosa, CA, USA) |
| Transmitting coil | 100 Coils | MCI-NRI [1] (Zegrze Poludniowe, Poland) |
| Receiving coil | 200 Coils | MCI-NRI [1] (Zegrze Poludniowe, Poland) |

[1] Made for the purpose of research at the Military Communication Institute—National Research Institute (MCI-NRI).

Due to the short-range nature of RFID communication, not exceeding a few centimeters, the measurements were carried out for two distances d between the transmitting coil and the RFID card of 5 and 3 cm, respectively.

For distances of $d = 5$ cm, the results obtained are shown in Table 2 and Figure 11. For distances of $d = 3$ cm, Table 3 and Figure 12.

**Table 2.** The value of the modulation depth factor as a function of the intensity of the interacting magnetic field for the distance $d = 5$ cm.

| Generator Power [dBm] | Magnetic Field [dBµA/m] | $U_{max}$ [mV] (Figure 7) | $U_{min}$ [mV] (Figure 7) | Modulation Depth Factor m |
|---|---|---|---|---|
| 36 | 123.6 | 345 | 227 | 0.21 |
| 35 | 123.6 | 342 | 225 | 0.21 |
| 34 | 122.8 | 317 | 215 | 0.19 |
| 33 | 121.9 | 282 | 202 | 0.16 |
| 32 | 120.9 | 257 | 192 | 0.14 |
| 31 | 119.9 | 225 | 175 | 0.13 |
| 30 | 119 | 201 | 165 | 0.10 |
| 29 | 118 | 173 | 150 | 0.07 |
| 28 | 117 | 155 | 138 | 0.06 |
| 27 | 116 | 140 | 127 | 0.05 |
| 26 | 115 | 127 | 116 | 0.05 |
| 25 | 114 | 112 | 104 | 0.04 |
| 24 | 113 | 99 | 95 | 0.02 |

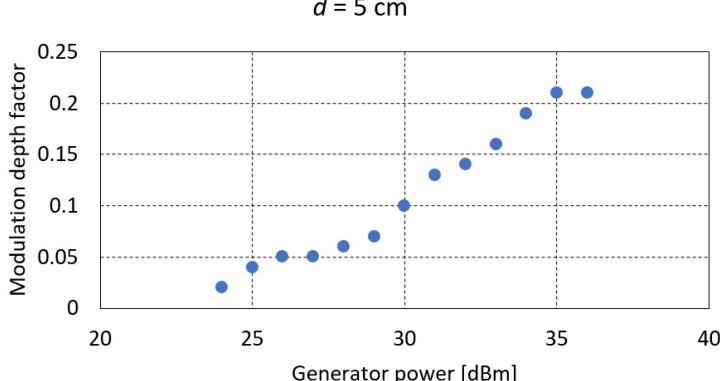

**Figure 11.** The value of the modulation depth factor as a function of generator power for the distance $d = 5$ cm.

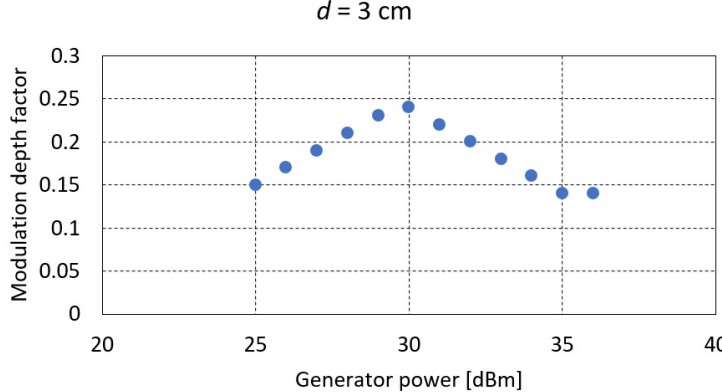

**Figure 12.** The value of the modulation depth factor as a function of generator power for the distance $d = 3$ cm.

**Table 3.** The value of the modulation depth factor as a function of the intensity of the interacting magnetic field for the distance $d = 3$ cm.

| Generator Power [dBm] | Magnetic Field [dBμA/m] | $U_{max}$ [mV] (Figure 7) | $U_{min}$ [mV] (Figure 7) | Modulation Depth Factor m |
|---|---|---|---|---|
| 36 | 126.7 | 482 | 362 | 0.14 |
| 35 | 126.6 | 477 | 362 | 0.14 |
| 34 | 125.9 | 462 | 335 | 0.16 |
| 33 | 125 | 442 | 310 | 0.18 |
| 32 | 124 | 422 | 282 | 0.20 |
| 31 | 123 | 402 | 257 | 0.22 |
| 30 | 122 | 382 | 235 | 0.24 |
| 29 | 121.1 | 357 | 222 | 0.23 |
| 28 | 120.1 | 318 | 205 | 0.21 |
| 27 | 119 | 285 | 192 | 0.19 |
| 26 | 118.1 | 253 | 180 | 0.17 |
| 25 | 117.1 | 230 | 170 | 0.15 |

Based on the measurements carried out, it was found that regardless of the level of signal strength from the generator and thus the value of the magnetic field strength, the maximum value of the modulation depth factor $m$ of the signal generated by the RFID card does not exceed 25%. This is related to the principle of operation of the card itself, which is not a generator in itself but only by keying the terminals of its coil (a magnetic antenna) affects the value of the voltage induced in the receiving coil (the phenomenon of mutual inductance). Further increasing the power of the generator (the magnetic field) makes the impact of the field directly from the transmitting coil decisive for the receiving coil. In this case, despite the increasing value of the voltage induced at the receiving coil terminals, the modulation depth factor $m$ decreases.

## 4. Improve Signal Quality and Extend Communication Range

Due to the declared near-range nature of communication between the RFID card and the reader [24] (during the experiments carried out in the laboratory, the maximum communication range $d$ was determined to be no more than 10 cm), attempts were made to receive signals from the card at longer distances, which was aimed at determining the real safety of this solution from the point of view of electromagnetic information penetration.

For this purpose, measurements were carried out using several additional electrical (electronic) systems, allowing, on the one hand, to improve the quality of the signal (increasing the modulation depth factor $m$) and, on the other hand, to increase the power (range) of the modulated signal or its distinctive features, enabling the reproduction of information sent by the RFID card [25].

First of all, the solution used in card readers was a resonant system tuned to the frequency of the RFID card. In this system, the inductor element is the receiving coil itself, so it was sufficient to expand the receiving coil system by connecting a capacitor in series to it (Figure 13).

The resonant frequency of such a system was 122.5 kHz, which ensured the correct operation of the RFID card (Figure 14), and a signal of this frequency was fed from the generator to the transmitting coil. The distance between the receiving coil and the RFID tag equal to 10 cm was experimentally selected in such a way as to obtain the optimal response of the receiving system in the entire tested range of the generator's output power (the power range is the same as in the case of modulation depth factor measurements).

The obtained measurement results for measurements without a resonant system are presented in Table 4 and Figure 15.

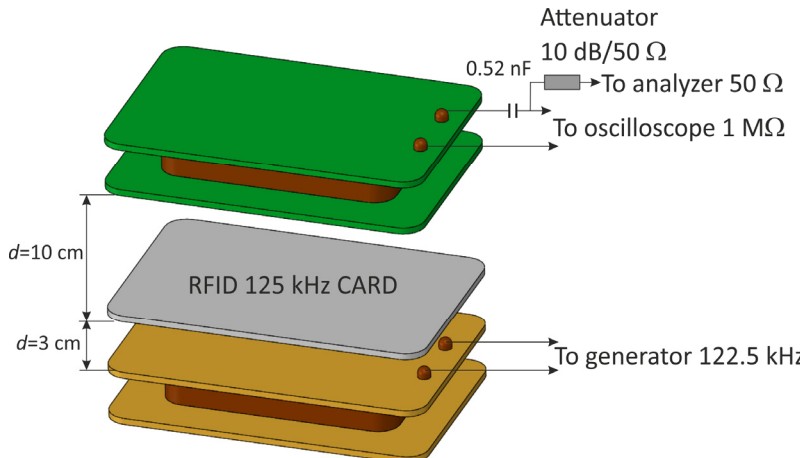

**Figure 13.** The measuring system for determining the influence of resonance on the modulation depth factor and signal amplitude.

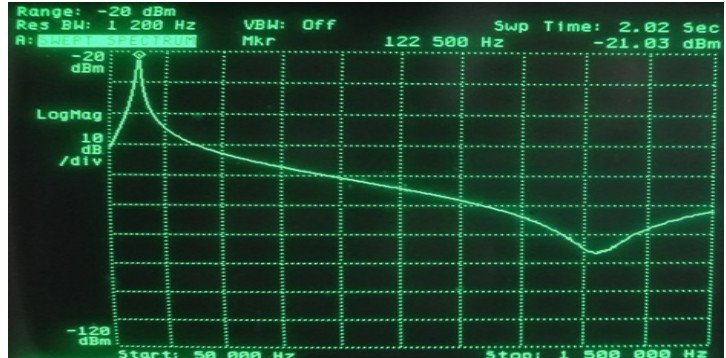

**Figure 14.** Determination of the resonant frequency of the system (122.5 kHz) using a spectrum analyzer.

**Table 4.** The value of the modulation depth factor as a function of the intensity of the interacting magnetic field for the distance $d = 3$ cm (comparison with and without a resonant circuit).

| Generator Power [dBm] | With Resonance | | | Without Resonance | | |
|---|---|---|---|---|---|---|
| | $U_{min}$ [mV] | $U_{min}$ [mV] | Modulation Depth Factor m | $U_{max}$ [mV] | $U_{min}$ [mV] | Modulation Depth Factor m |
| 35 | 685 | 635 | 0.04 | 122 | 108 | 0.06 |
| 34 | 640 | 582 | 0.05 | 116 | 99 | 0.08 |
| 33 | 585 | 525 | 0.05 | 107 | 91 | 0.08 |
| 32 | 535 | 470 | 0.06 | 101 | 81 | 0.11 |
| 31 | 492 | 422 | 0.08 | 95 | 75 | 0.12 |
| 30 | 452 | 380 | 0.09 | 88 | 66 | 0.14 |
| 29 | 422 | 342 | 0.10 | 84 | 62 | 0.15 |
| 28 | 382 | 315 | 0.10 | 76 | 57 | 0.14 |
| 27 | 339 | 279 | 0.10 | 69 | 51 | 0.15 |
| 26 | 304 | 255 | 0.09 | 64 | 50 | 0.12 |
| 25 | 271 | 233 | 0.07 | 58 | 47 | 0.10 |
| 24 | 241 | 212 | 0.06 | 52 | 44 | 0.08 |
| 23 | 219 | 194 | 0.06 | 47 | 41 | 0.07 |
| 22 | 219 | 194 | 0.06 | 47 | 41 | 0.07 |
| 21 | 196 | 177 | 0.05 | 43 | 38 | 0.06 |
| 20 | 175 | 164 | 0.03 | 39 | 35 | 0.05 |

Based on the obtained results, it was found that the amplitude parameters (signal strength determining the communication range) of the signal have improved significantly. An approximately five-fold increase in signal levels was observed. At the same time, this affected the communication range, which for a system with resonance increased to about 15 cm compared to 10 cm for a case without resonance.

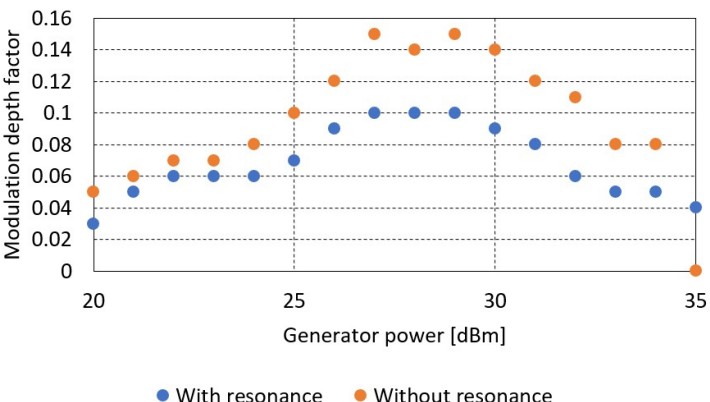

**Figure 15.** The modulation depth factor as a function of the generator signal power delivered to the transmitting coil for the case with and without the use of a resonant system.

The disadvantage of this solution, however, is the reduction of the modulation depth factor. In extreme cases, from 0.15 to 0.1, the signal detection capabilities were significantly affected, and the communication range was reduced.

The obtained results confirmed the possible phenomenon. The resonant system amplifies the carrier wave of the resonant frequency to the greatest extent (increasing the amplitude of the observed signals), which by itself does not carry information about the transmitted data. The remaining components of the spectrum are either amplified to a lesser extent or remain unchanged, resulting in a reduction in the depth factor (Figure 16).

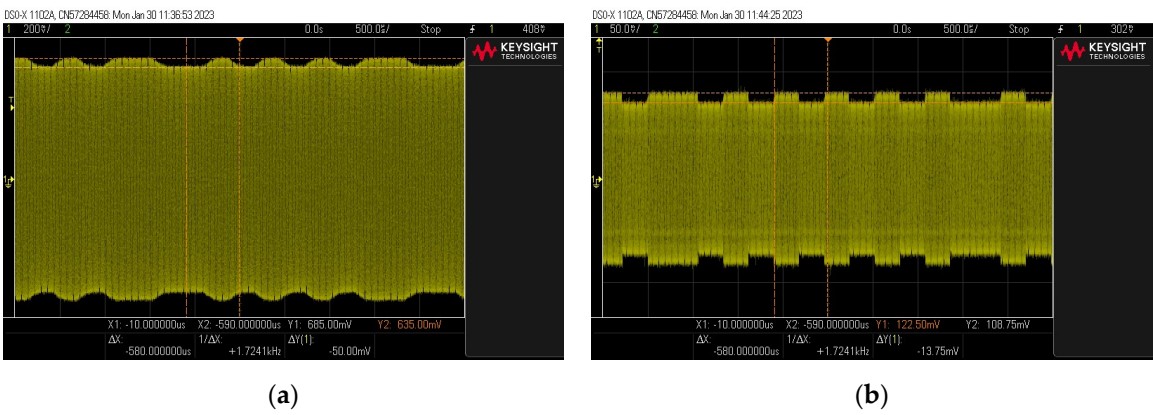

(**a**)                                                (**b**)

**Figure 16.** Examples of signal waveforms for the case using (**a**) and without the use of (**b**) a resonant system.

The improvement of signal quality, i.e., the increase in the modulation depth factor *m*, should be influenced by the opposite effect—attenuation of the carrier wave and amplification of other components of the signal spectrum. Of course, on the other hand, this may have an adverse effect on the amplitude parameters of the signal, i.e., at the same time reducing the range.

This phenomenon was confirmed in subsequent tests carried out in the measurement system presented in Figure 17, for which the resonant frequency of the filter was 129.5 kHz (Figure 18), which at the same time ensured the correct operation of the RFID card. As in the previous case, the distance between the receiving coil and the RFID card was experimentally selected in such a way as to obtain the optimal response of the receiving system over the entire test range of the generator's output power (the power range as in the case of modulation depth factor measurements).

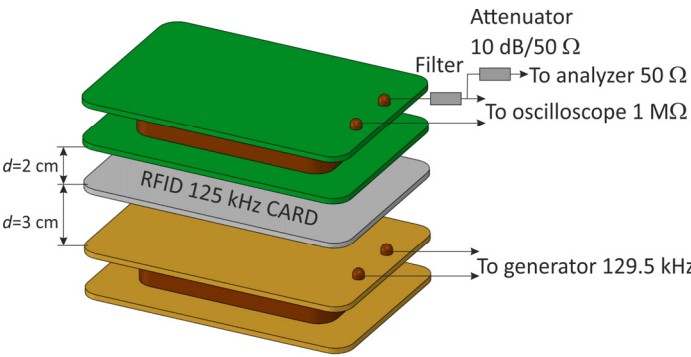

**Figure 17.** Measuring system for determining the influence of a band-stop resonant filter on the modulation depth factor and signal amplitude.

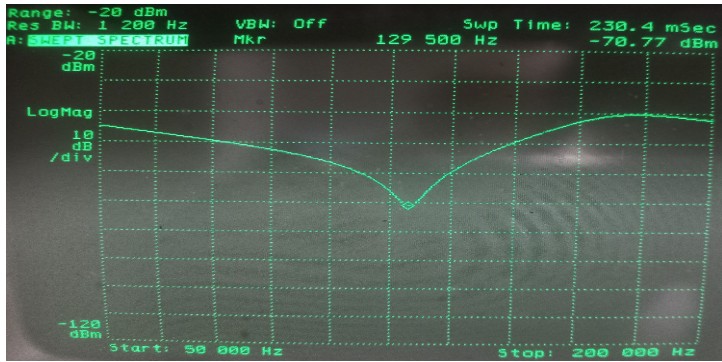

**Figure 18.** Determination of the resonant frequency of the filter (129.5 kHz) by using a spectrum analyzer.

The results obtained during the measurements showed a significant reduction in the signal amplitude with respect to the system without a filter (Figure 19). There was even a fifteen-fold decrease in the amplitude value. As a result, the communication range has also been significantly reduced—to about 5 cm. At the same time, an improvement in the modulation depth factor *m* was observed, which reached a value of about 20%.

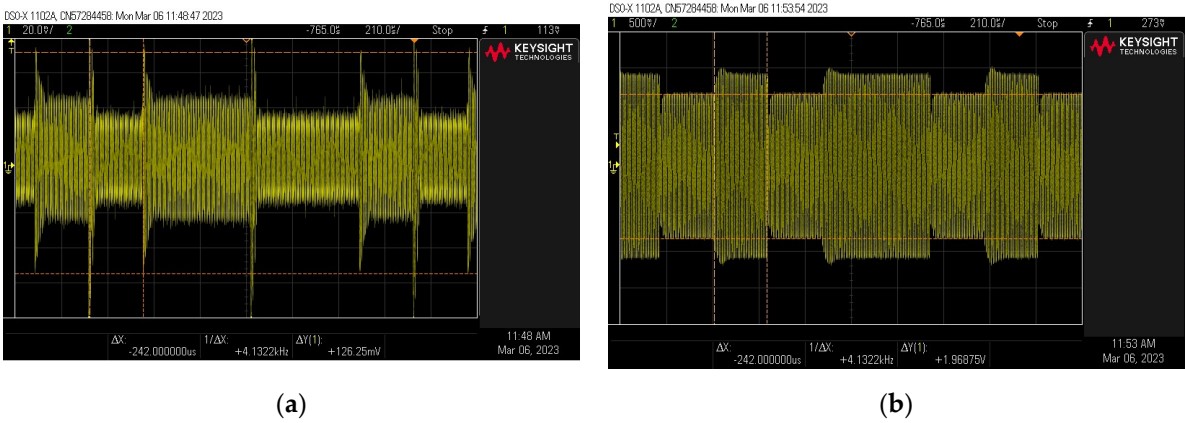

(a)  (b)

**Figure 19.** Examples of signal waveforms for the case using (**a**) and without the use of (**b**) a resonant filtering system.

The last tested solution to enable reading data from an RFID card from a greater distance than declared by the manufacturers was a resonant active filter system with an amplifier. The task of such a system was to suppress the carrier wave while strengthening the remaining part of the signal spectrum. The filter is built with a dual operational amplifier (AD8608). A resonant band-stop filter is installed in the feedback circuit. The

second stage of the amplifier was used as an ordinary amplifier, operating in the entire useful frequency band. A diagram of this system is shown in Figure 20.

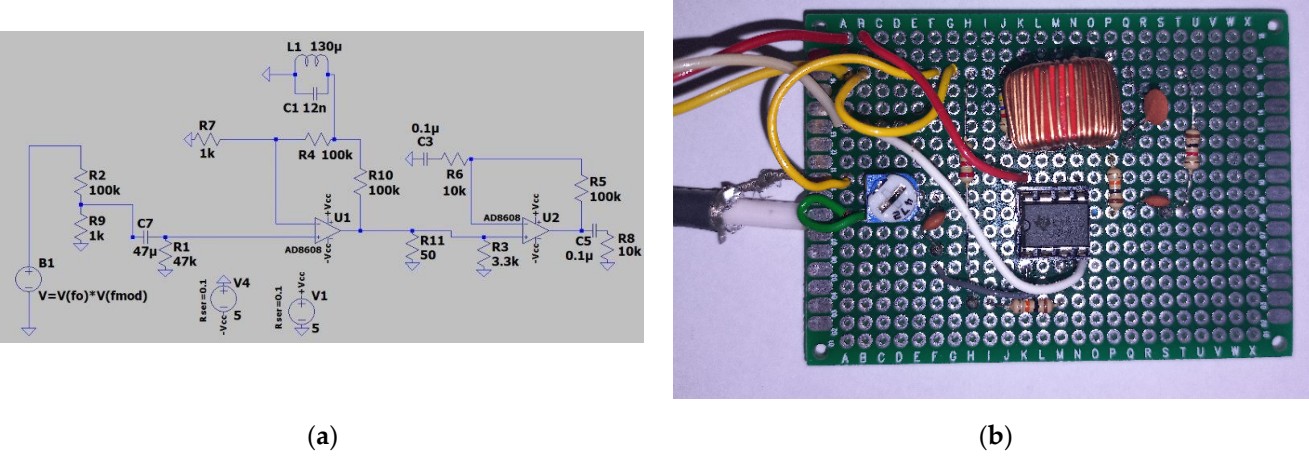

(**a**)  (**b**)

**Figure 20.** (**a**) Diagram of the active band–stop filter; (**b**) practical implementation of the active band–stop filter.

The filter made in this way was connected to the terminals of the receiving coil in the system shown in Figure 21.

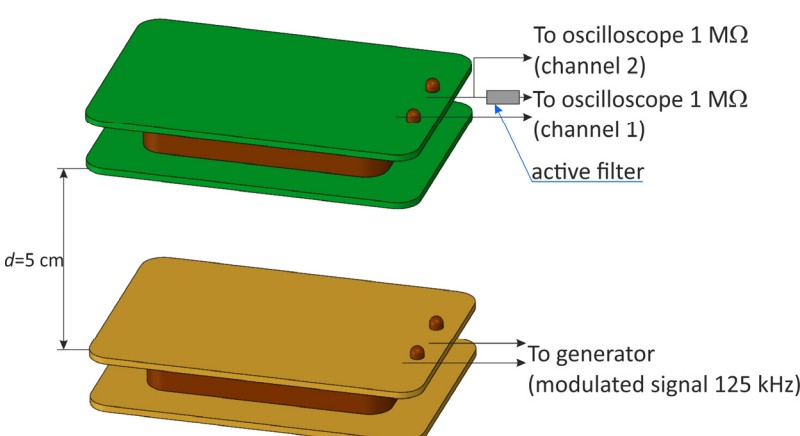

**Figure 21.** Measuring system for checking the functioning of the active band–stop filter.

First, measurements were carried out on a system without an RFID card. They were aimed at assessing the functioning of the filter. For this purpose, instead of an RFID card, an amplitude-modulated signal from the generator was sent directly to the transmitting coil. The frequency of the signal (carrier wave) was $f_n = 125$ kHz, and the modulating signal was a rectangular signal with a frequency of $f_p = 1$ kHz. Measurements were carried out for several values of the modulation depth factor $m$ while observing the signal before and after the filter. Adjustment of the modulation depth factor was achieved by means of appropriate modulation parameter settings in the signal generator. The m-factor for the signal measured upstream of the filter was determined using an oscilloscope (the generator readings were only an approximate value).

As expected, the filter did not adversely affect the amplitude of the $U$ signal but at the same time significantly improved the readability of the signal—a large increase in the modulation depth factor $m$, as shown in Figure 22. In the case of signals with an almost zero modulation depth factor $m$ measured before the filter, the signal became much clearer after the filter (the modulation factor increased to a value of approximately $m = 10\%$). In

the case of the source signal with a modulation depth factor of $m \approx 3\%$, the output signal $U$ was characterized by a modulation depth factor of $m \approx 40\%$.

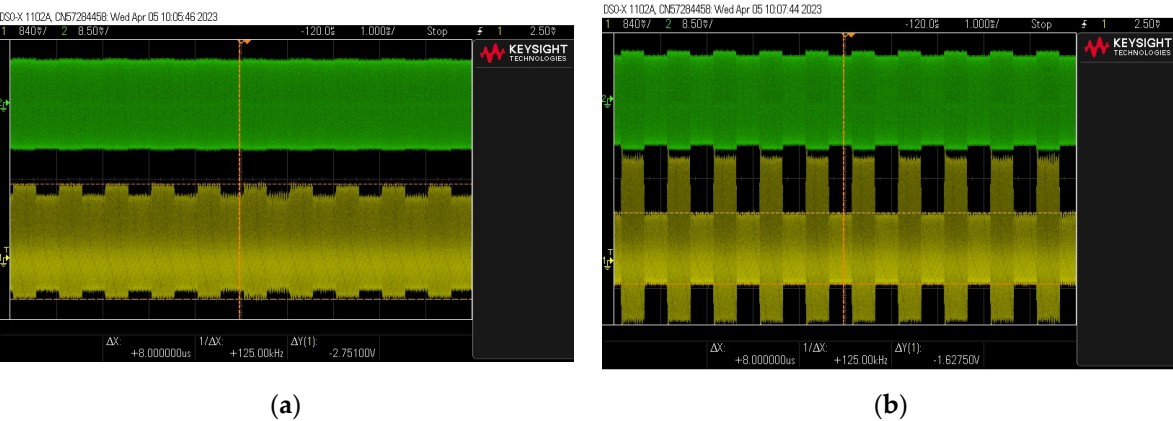

(**a**)
(**b**)

**Figure 22.** Examples of signal waveforms before (green) and after the filter (yellow) for different input signal modulation factors: (**a**) the case of a source signal with an almost zero modulation depth factor $m$; (**b**) the case of a source signal with modulation depth factor $m \approx 3\%$.

After verifying the correct functioning of the filter, measurements were carried out using an RFID card. The measurements were carried out in the system, as shown in Figure 23.

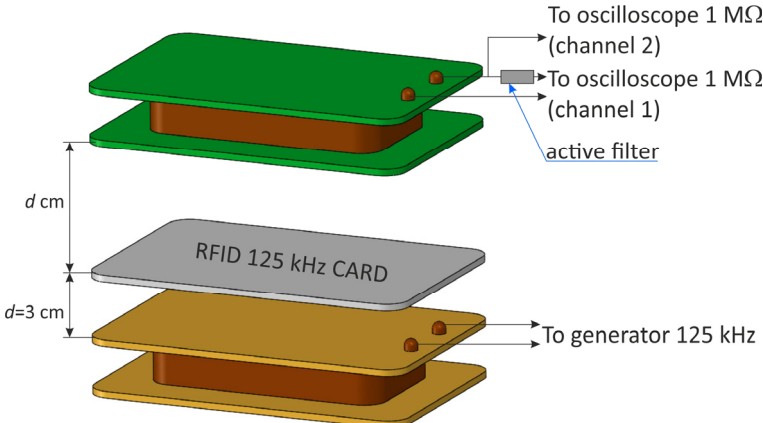

**Figure 23.** Measurement system for determining the effectiveness of an active band-stop filter in a system with an RFID card.

During the measurements, the distance $d$ was chosen experimentally so that the $U$ signal before the active filter was characterized by a very small modulation depth factor $m = 1\%$. It was observed to what extent the signal behind the filter was improving.

The results of this experiment are shown in Figure 24. They show that even in the case of signals with very small values of the modulation depth factor, for which the signal seems illegible, the signals behind the active filter are of much better quality. The signal with a modulation factor $m \approx 1\%$ (practically illegible) behind the filter is characterized by a factor of $m \approx 3\%$, which allows its effective reading (decoding). The filter works even more advantageously in the case of modulation depth factors greater than $m \approx 1\%$. For a signal with a $m \approx 3\%$ ratio, the output signal has a factor of $m \approx 13\%$.

Despite the high effectiveness of this solution, the maximum communication range in this case also does not exceed several centimeters.

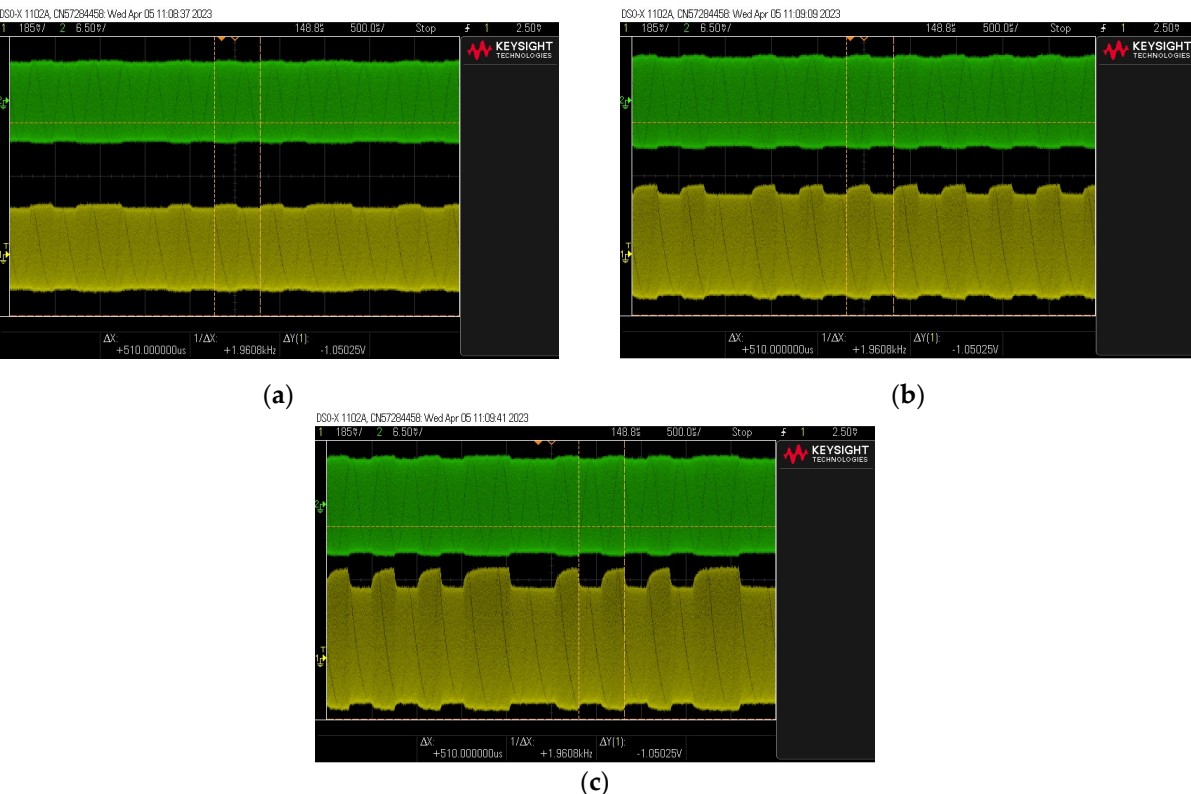

**Figure 24.** Examples of signal waveforms before (green) and after (yellow) filter for three values of modulation depth factor before filter: (**a**) $m = 1$, (**b**) $m = 3$, and (**c**) $m = 5$.

## 5. Electromagnetic Safety of Access Control Systems

The experiments described so far, aimed at increasing the communication range between the RFID card and the receiver, did not allow for a significant increase in the distance between the receiver and the card. Even with the use of an active filter, the communication range was still close to that declared by RFID system manufacturers.

Therefore, the entire access control system was tested, which was built from the following elements:

- VC-1200 controller;
- Power supply;
- Battery (power backup system);
- Two R101EM RFID card readers;
- Electric strike door.

The diagram of this system and its practical implementation are presented in Figure 25.

First of all, it was checked in which components of the access control system the signal from the RFID card may be present. It was found that already on the signal lines between the reader and the controller, a signal is transmitted that cannot be identified in the aspect of electromagnetic information security. After two data lines (Figure 26), pulses responsible for logical "1" (line D1) and logical "0" (line D0) are transmitted. Attempts to receive such information electromagnetically will always give one of two results: either the string "00000..." or the string "11111...", regardless of the signal transmitted by the card (summation of signals from two lines and no possibility to determine which signal comes from which line).

Therefore, the only element that can emit a signal from the card is the card reader. In the next stage, research was carried out to reproduce the data transmitted by the RFID card. The measuring apparatus listed in Table 5 was used for the measurements.

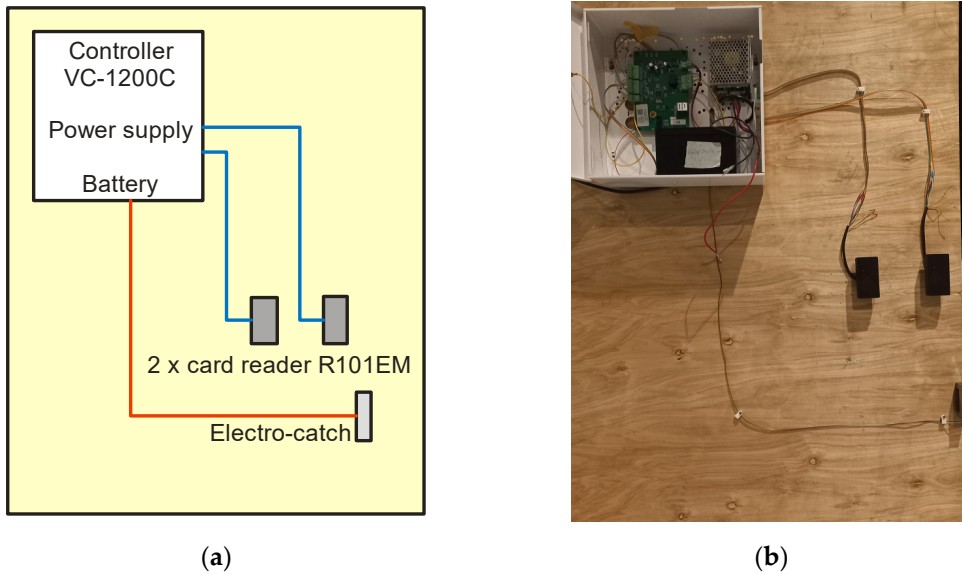

(**a**)  (**b**)

**Figure 25.** Access control system (ACS): (**a**) diagram and (**b**) practical implementation.

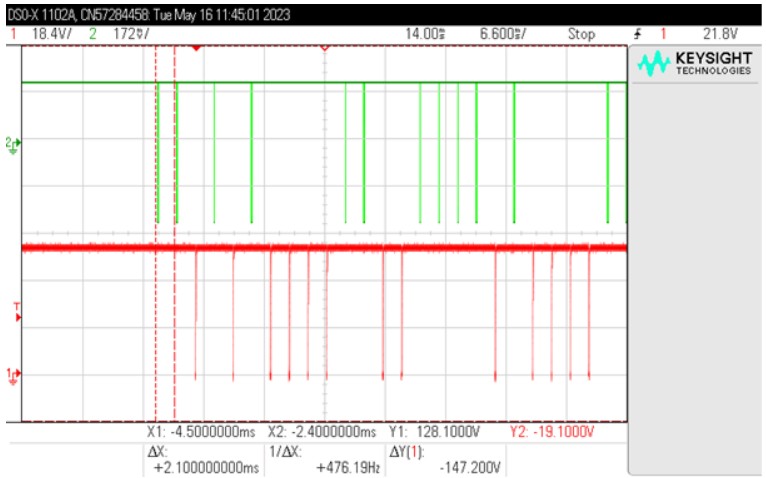

**Figure 26.** Simultaneous reading of signals from lines D0 (green) and D1 (red).

**Table 5.** Measuring equipment used to receive signals from a complete access control system.

| Equipment | Type | Producer |
|---|---|---|
| Test receiver | FSWT26 | Rohde and Schwarz |
| Oscilloscope | DSO90404A | Agilent (Santa Clara, CA, USA) |
| Rod antenna | HE525 | Rohde and Schwarz |
| Measurement computer | Pentium i7 | HP |

The tests were carried out in an anechoic shielding cabin in the measuring system shown in Figure 27.

Due to the low frequency range of communication, the measurements were limited to the frequency range from 10 kHz to 50 MHz. During the tests, the access system worked only on battery power. The power supply used in the system was characterized by undesirable electromagnetic emissions at a high level, precisely at the frequencies of RFID communication, so it was not possible to filter disturbances from the power supply as described in [26], and connecting the 230 V mains supply caused the emission of disturbances at a level that made it impossible to receive the searched signals [26,27], as shown in Figure 28.

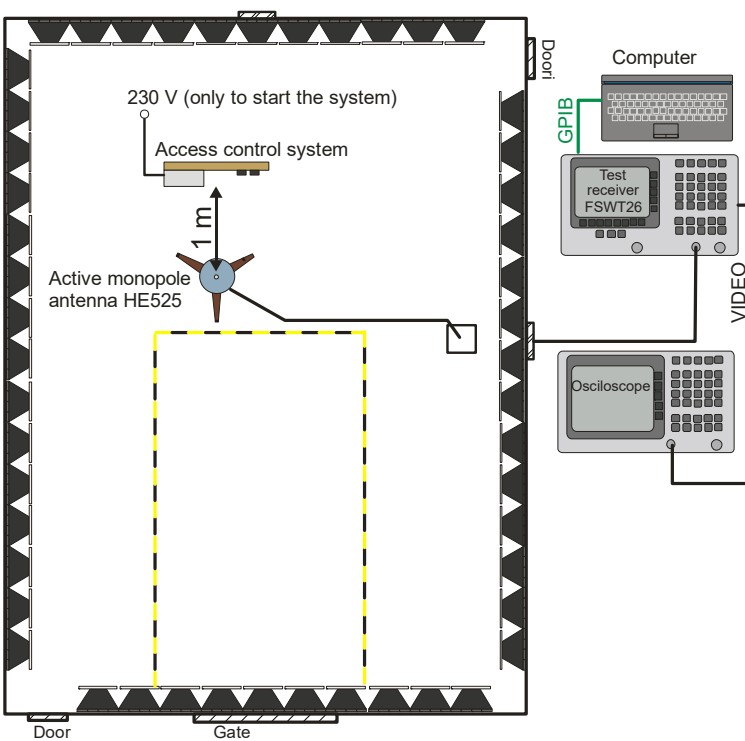

**Figure 27.** Measurement system for identifying data transmitted by the RFID card.

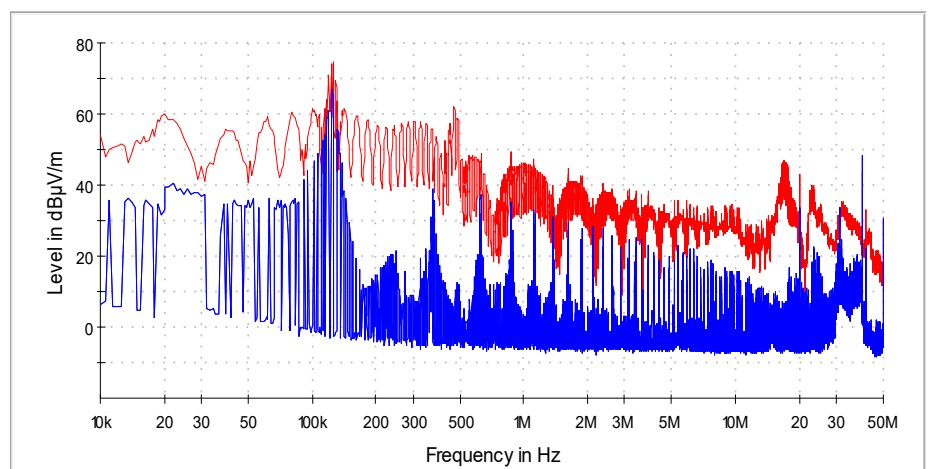

**Figure 28.** Measured level of electromagnetic disturbance from the ACS in the case of power supply from the power supply 230 V (red) and 12 V battery (blue).

Based on the analysis of the obtained measurement results, several frequencies were selected for further measurements, for which there was a high probability of receiving the desired signals. This was the nominal frequency of the card reader $f_0 = 125$ kHz and the first few harmonics, $f_1 = 250$ kHz, $f_2 = 375$ kHz, $f_3 = 500$ kHz, and $f_4 = 625$ kHz.

After tuning the receiver to the required frequency, a demodulated signal from the video output of the measuring receiver on the oscilloscope was observed.

Only for the frequency $f_0 = 125$ kHz it was possible to receive signals correlated with the data transmitted by the RFID card. In this case (with the use of the specified apparatus), the maximum distance from which it was possible to identify the transmitted information did not exceed $d = 2m$.

Examples of reproduced signal waveforms for several types of data stored on RFID cards are shown in Figure 29.

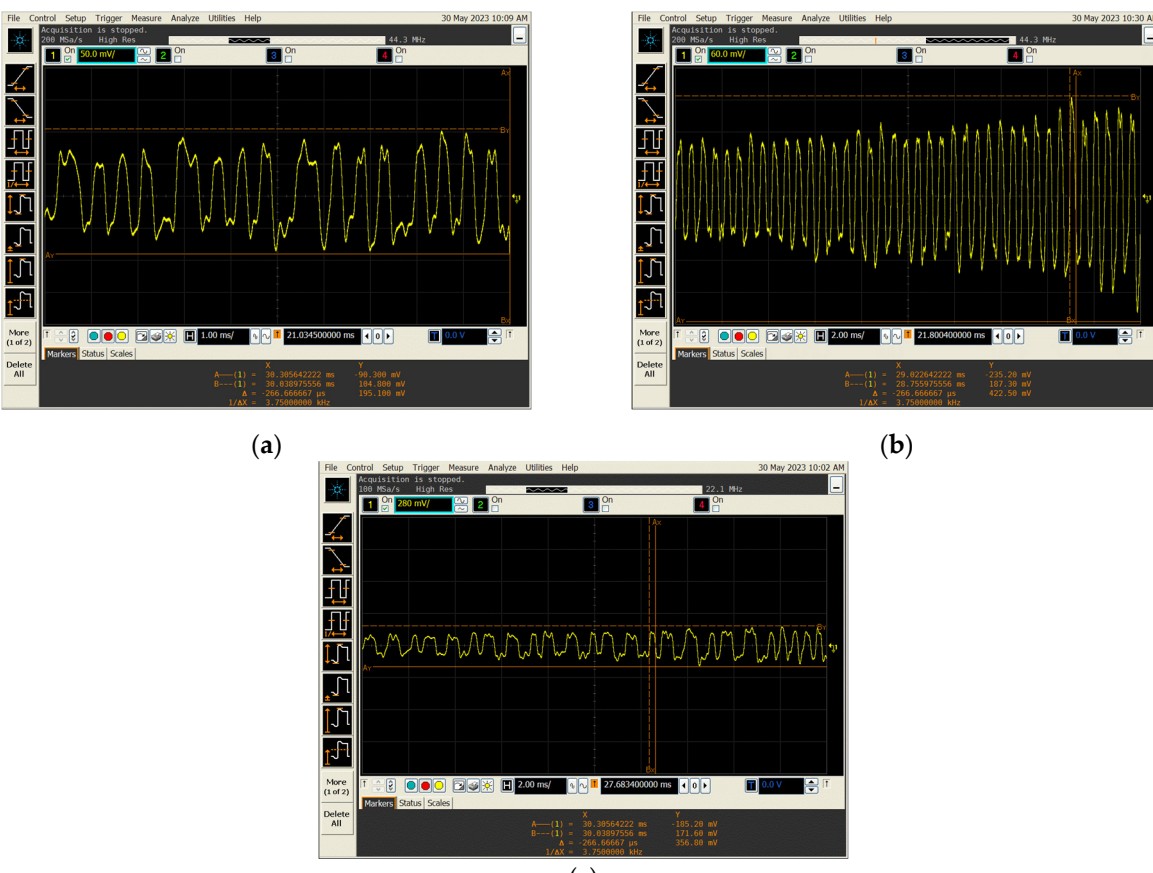

**Figure 29.** Exemplary reconstructed signal waveforms correspond to the data transmitted by the RFID card. (**a**) Transmission of the data string "1111..."; (**b**) transmission of the data string "0000..."; and (**c**) transmission of pre-programmed data—a string of almost random data.

It should be noted that the waveforms read are encoded using Manchester encoding, i.e., in the same form as the information transmitted by the RFID card.

In the case of electromagnetic safety tests of a complete access system, one more important element of this system, i.e., the computer (server), was not taken into account. This computer, when connected to the controller, is used both to program access permissions and to read and write individual events, such as entering and leaving the zone or attempting unauthorized access. However, this element of the system constitutes ICT security (it is important from the point of view of resistance to hacker attacks), and for this reason it was not subject to the described tests.

## 6. Conclusions and Future Works

The article discusses the issue of the electromagnetic safety of RFID systems, i.e., the possibility of non-invasive acquisition of data stored on the cards of this system. This issue is important because such cards can be used in time registration systems (for professional activity), entry to restricted zones in workplaces and offices, or they can contain warehouse data. Copying information from such a card may allow unauthorized access to rooms and sensitive data.

The tests were carried out on a simulation system and a real system equipped with a card reader and the entire control system. The obtained test results confirmed the safety of the tested solution. The communication range between the card and the reader declared by the manufacturers of RFID systems is several cm and is consistent with the results obtained during the tests. The attempts to increase the reach of communication described in the article did not affect him in a significant way. The maximum distances from which it was

possible to read the data did not exceed several cm. Therefore, communication between the card and the reader can be considered safe. Unauthorized persons are not able to restore the data stored on the card and possibly copy such a card without direct access to it.

However, considering the RFID system as a whole and also taking into account card readers, it is not completely secure. The principle of operation of the system requires bringing the card close to the reader, which causes it to start. At this point, it becomes a source of significant electromagnetic emissions. These emissions allow the identification of data stored on the card in the form of logical "0" and "1" and their reproduction. Registration of these electromagnetic emissions is possible even from a distance of 2 m. Considering the aspects of the electromagnetic infiltration process, the distance of 2 m between the receiving antenna and the emission source is not a small distance. Previous studies of ICT (Information and Communication Technologies) devices processing data in graphic form conducted by the authors of the article show that the process of electromagnetic infiltration can be carried out even from a distance of 1 m.

Further work in the area of electromagnetic security in RFID systems will focus on the issues of protecting card readers against electromagnetic infiltration. In addition, electromagnetic safety will be analyzed for systems operating at higher frequencies. Further work in the area of electromagnetic security in RFID systems will focus on the issues of protecting card readers against electromagnetic infiltration. In addition, electromagnetic safety will be analyzed for systems operating at higher frequencies.

At the same time, it should be noted that the presented research results concern only RFIF LF systems, i.e., operating at a frequency of 125 kHz. Due to significant differences in relation to other RFID systems (RFID, HF, and UHF), the presented results cannot be applied to them. Studies of the electromagnetic security of other systems, including in particular RFID HF (13.56 MHz) used in payment systems, will be the subject of further research by the authors.

**Author Contributions:** Conceptualization, S.M., T.D. and A.F.; methodology, S.M. and A.F.; validation, S.M., A.F. and I.K.; formal analysis, S.M. and I.K.; investigation, S.M. and A.F.; resources, I.K. and S.M.; data curation, S.M. and A.F.; writing—original draft preparation, S.M.; writing—review and editing, A.F.; visualization, T.D.; supervision, I.K.; project administration, S.M.; funding acquisition, I.K. All authors have read and agreed to the published version of the manuscript.

**Funding:** This research received no external funding.

**Data Availability Statement:** Not applicable.

**Conflicts of Interest:** The authors declare no conflict of interest.

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
