# Peer review of "Electromagnetic Safety of Short-Range Radio Frequency Identification Systems"

_electronics, doi:10.3390/electronics12214391_

Round 1

Reviewer 1 Report

This article presents an experimental  study of the electromagnetic safety of RFID systems, i.e. the possibility of non-invasive acquisition of data stored on the cards of this system. The goal is to verify the security of the entry in restricted zones. Copying information frum such cards may allow unauthorized access in sensitive areas.

The paper is well-written, very well illustrated (28 figures), the experiments are well explained. The obtained results are interesting and should be known by the concerned researchers. Therefore, I suggest the publication of this contribution.

Author Response

Dear Reviewer

Thank you for providing the detailed and constructive review for our paper which allowed to increase a scientific level of this paper. We tried to reply for each point mentioned in the review. Our corrections were highlighted in manuscript by blue and red colours. We hope that the new version of our manuscript looks better and meets your requirements.

Here we’d like to reply for each point mentioned in the review:

Does the introduction provide sufficient background and include all relevant references?

Can be improved

Our response

We agree with your opinion. We have improved the abstract according to the given guidelines: We have improved its informative function, extended it and hopefully made it more attractive to readers.

Specifically, we added the following text in the abstract, please see lines 20 to 26:

“Particular attention has been paid to the possibility of unauthorized acquisition of information contained in the identification card in order to, for example, copy it and gain access to specific protected zones. An analysis of the security of such systems was carried out not only in relation to the data carriers themselves, but also to complete access control systems installed in buildings. The research focused especially on the ability to determine the range of information penetration, i.e. the distance of remote information acquisition using electromagnetic radiated emissions”.

 According to your suggestion, we revised the manuscript from the abstract to the conclusions. We add new abstract and defined more precisely motivation and goals. The introduction has been greatly expanded to detail the objectives of the research.

Specifically, we added the following text in the introduction, please see lines 57 to 79, the Figure 1 was also added:

“The above examples of RFID systems application show the possibilities of using the technology in various areas of life. The authors of the publication do not focus on data protection issues. Often, however, RFID systems process information that can determine the security not only of persons, but also of companies or the state. Hence, the authors of the article attempted to conduct comprehensive research and analysis of the obtained results in terms of the possibility of carrying out the process of electromagnetic infiltration of RFID systems. If the systems were found to be immune to electromagnetic eavesdropping, scenarios that could increase the effectiveness of RFID systems as sources of unwanted electromagnetic emissions were analyzed.”

“The aim of the study was to determine the possibility of increasing the range of radio communication to a value that allows unauthorized persons to read the data from the identifier. For this purpose, various electronic circuits were used to increase the level of the signal emitted by the identifier, to improve the signal quality by increasing the modulation depth factor, and to analyze the signals emitted by the complete access system when the identifier is read. In order to reduce undesirable electromagnetic disturbances that may come from the elements of the access system and may have an adverse effect on the measurement results, magnetic coils (receiving and transmitting) of our own design were used, cooperating with measuring equipment with low levels of internal electromagnetic disturbances. In addition, the measurements were carried out in an anechoic chamber, which ensured separation from the external electromagnetic environment (Figure 1)”

In addition to the above comments, we conducted extensive editing of English language and style. We realize that, despite all our efforts, this article may not be language and style error-free. Therefore, the final version, after possible acceptance, will be corrected by the proofreading service in the MDPI.

We look forward to hearing from you in due time regarding our submission and to responding to any further questions and comments you may have.

Best Regards

Authors

Reviewer 2 Report

1. The abstract is quite imperfect, is short and isn't informative. Please be concise; that should be attractive to the readers.

2. The introduction is too short. Dos not contain the aims of the reserch.

3. The references are relevant. But, this part needs to be fully developed. Please come out with a systematic literature review.

4. Please develop part of the methodology. I need to understand how you could measurement the data, why you chose the methods, and how you could develop your measurement system. The writing is confusing to the readers. However, the analysis part needs more clarification. Modeling section needs to be explained. The steps of the proposed model and method are very simple, the steps are not explained enough.

5. I consider it necessary to make a critical comparison with existing, similar results, thereby highlighting and supporting the advantages of the presented reserch's result.

6. At first reading, it is difficult to follow the thought the mesurement reseults , it should be specified in order to avoid misunderstandings.

7. Discussion on Limitations of the Methodology: The paper does not discusses the limitations of the methodology, an important section as it helps readers understand the limitations of the research. However, a more detailed explanation of how these limitations impact the research findings, and how these limitations can be addressed in future research might be needed.

Author Response

Dear Reviewer

Thanks for providing the detailed and constructive review for our paper which allowed to increase a scientific level of this paper. We tried to reply for each point mentioned in the review. Our corrections were highlighted in manuscript by blue and red colours. We hope that the new version of our manuscript looks better and meets your requirements.

Here we’d like to reply for each point mentioned in the review:

  1. The abstract is quite imperfect, is short and isn't informative. Please be concise; that should be attractive to the readers

We have improved the abstract according to the given guidelines: We have improved its informative function, extended it and hopefully made it more attractive to readers.

Specifically, we added the following text in the abstract, please see lines 20 to 26:

“Particular attention has been paid to the possibility of unauthorized acquisition of information contained in the identification card in order to, for example, copy it and gain access to specific protected zones. An analysis of the security of such systems was carried out not only in relation to the data carriers themselves, but also to complete access control systems installed in buildings. The research focused especially on the ability to determine the range of information penetration, i.e. the distance of remote information acquisition using electromagnetic radiated emissions”.

  1. The introduction is too short. Dos not contain the aims of the research.

Our response

We agree with your opinion. According to your suggestion, we revised the manuscript from the abstract to the conclusions. We add new abstract and defined more precisely motivation and goals.The introduction has been greatly expanded to detail the objectives of the research.

Specifically, we added the following text in the abstract, please see lines 57 to 79, the  Figure 1 was also added:

“The above examples of RFID systems application show the possibilities of using the technology in various areas of life. The authors of the publication do not focus on data protection issues. Often, however, RFID systems process information that can determine the security not only of persons, but also of companies or the state. Hence, the authors of the article attempted to conduct comprehensive research and analysis of the obtained results in terms of the possibility of carrying out the process of electromagnetic infiltration of RFID systems. If the systems were found to be immune to electromagnetic eavesdropping, scenarios that could increase the effectiveness of RFID systems as sources of unwanted electromagnetic emissions were analyzed.”

“The aim of the study was to determine the possibility of increasing the range of radio communication to a value that allows unauthorized persons to read the data from the identifier. For this purpose, various electronic circuits were used to increase the level of the signal emitted by the identifier, to improve the signal quality by increasing the modulation depth factor, and to analyze the signals emitted by the complete access system when the identifier is read. In order to reduce undesirable electromagnetic disturbances that may come from the elements of the access system and may have an adverse effect on the measurement results, magnetic coils (receiving and transmitting) of our own design were used, cooperating with measuring equipment with low levels of internal electromagnetic disturbances. In addition, the measurements were carried out in a non-reflective shielding booth, which ensured separation from the external electromagnetic environment (Figure 1)”

  1. The references are relevant. But, this part needs to be fully developed. Please come out with a systematic literature review.

Our response

The article is original and was written after reviewing the listed literature and references items. In our opinion, matching literature references items and to the finished article is counterproductive.

But we added text between lines 136 to 137 which shows that values of depth factor determined during our measurements are similar to those presented in reference [18]:

“The values of the modulation depth factor determined during the measurements are similar to those presented in [18] for the case of the nonlinear transponder model.”

We added also text between lines 166 to 169 which shows similar behavior of electromagnetic field in the case of increasing the distance d  similarly like in reference [22]:

“During these measurements, as shown in [22], a very high attenuation of the electromagnetic field was noticeable in the case of increasing the distance d and a high sensitivity to changes in the angle of the coils in relation to each other.”

We've also added text referring to references [25, 26] between lines 378 to 383, which describes similar difficulties with our ability to filter out interference from the power system (Figure 28):

“The power supply used in the system was characterized by undesirable electromagnetic emissions at a high level, precisely at the frequencies of RFID communication, so it was not possible to filter disturbances from the power supply as described in [25] and connecting the 230 V mains supply caused the emission of disturbances at a level that made it impossible to receive the searched signals [25,  26], as shown in Figure 27.”

  1. Please develop part of the methodology. I need to understand how you could measurement the data, why you chose the methods, and how you could develop your measurement system. The writing is confusing to the readers. However, the analysis part needs more clarification. Modelling section needs to be explained. The steps of the proposed model and method are very simple, the steps are not explained enough.

Our response

We have tried to expand on the part of the manuscript  that deals in particular with the way of choosing measurement methods and how they are implemented. In particular, the part of the manuscript concerning Improve signal quality and extend communication range using the phenomenon of resonance and the use of band-pass filtration aimed at cutting out the components of the spectrum that do not carry information (carrier) and boosting the information components has been detailed.

For example between lines 149  to 155 we added text:

“As noted in the introduction of the article, the tested RFID system, due to the value of the operating frequency and areas of application, may not be susceptible to electromagnetic infiltration. However, just like in the case of IT systems that process information in graphic form (laser printers, computers, laptops, multifunctional devices), the RFID system can be subjected to external factors that change the operating parameters of electronic systems. In this way, additional sources of unwanted emissions are activated or the effectiveness of existing ones is increased.”

We also specified the main  goal of measurements (text between lines 156 to 160):

“The main objective of the measurements was to check whether increasing the magnetic field (the strength of the signal delivered to the transmitting coil) acting on the identifier would allow to increase the communication range. In this case, the communication range is the same as a sufficiently large value of the modulation depth factor necessary for the correct detection of the received signal.”

It was also explained how were selected  the resonant frequency of the system for correct operation of RFID tag and the distance between  the receiving coil and the RFID tag, (text between lines 223 to 228):

“The resonant frequency of such a system was 122.5 kHz, which ensured the correct operation of the RFID card (Figure 14) and a signal of this frequency was fed from the generator to the transmitting coil. The distance between the receiving coil and the RFID card equal to 10 cm was experimentally selected in such a way as to obtain the optimal response of the receiving system in the entire tested range of the generator's output power (the power range is the same as in the case of modulation depth factor measurements).”

And we also added text (between lines 268 to 271):

“As in the previous case, the distance between the receiving coil and the RFID card was experimentally selected in such a way as to obtain the optimal response of the receiving system over the entire test range of the generator's output power (the power range as in the case of modulation depth factor measurements).”

Between lines 306 and 309 a passage has been added on how to adjust the modulation depth factor:

“Adjustment of the modulation depth factor was achieved by means of appropriate modulation parameter settings in the signal generator. The m-factor for the signal measured upstream of the filter was determined using an oscilloscope (the generator readings were only an approximate value).”

  1. I consider it necessary to make a critical comparison with existing, similar results, thereby highlighting and supporting the advantages of the presented reserch's result

Our response

This comparison has been done as far as possible, as there are few studies in this area. Most of them were conducted in the 13.56 MHz frequency range.

  1. At first reading, it is difficult to follow the thought the measurement results , it should be specified in order to avoid misunderstandings.

Our response

Information on the results obtained and how they were achieved, as well as their relevance to the security of short-range communications, has been clarified, which we hope should prevent misunderstandings.

This overlaps in part with the replies to the comments in point 4, to which we have already responded (above).

Additionally we added text which explains the nature of RFID communication:

“Due to the short-range nature of RFID communication, not exceeding a few centimeters”

  1. Discussion on Limitations of the Methodology: The paper does not discusses the limitations of the methodology, an important section as it helps readers understand the limitations of the research. However, a more detailed explanation of how these limitations impact the research findings, and how these limitations can be addressed in future research might be needed.

Our response

A discussion on the limitations of the research methodology has been added, which should help the reader understand the limitations of the research methods used and how these limitations may affect the results of the research. This discussion was linked to the limitations of short-range communication.

For example we added text between lines 401 to 407:

“In the case of electromagnetic safety tests of a complete access system, one more important element of this system, i.e. the computer (server), was not taken into account. This computer, when connected to the controller, is used both to program access permissions and to read and write individual events, such as entering and leaving the zone or attempting unauthorized access. However, this element of the system constitutes ICT security (it is important from the point of view of resistance to hacker attacks) and for this reason it was not subject to the described tests.”

Also text between lines 437 to 439 was added:

“Previous studies of ICT devices processing data in graphic form conducted by the authors of the article show that the process of electromagnetic infiltration can be carried out even from a distance of 1 m.”

And text between lines 446 to 451 was added.

“At the same time, it should be noted that the presented research results concern only RFIF LF systems, i.e. operating at a frequency of 125 kHz. Due to significant differences in relation to other RFID systems (RFID, HF and UHF), the presented results cannot be applied to them. Studies of electromagnetic security of other systems, including in particular RFID HF (13.56 MHz) used, m.in, in payment systems, will be the subject of further research by the authors.”

In addition to the above comments, we conducted extensive editing of English language and style. We realize that, despite all our efforts, this article may not be language and style error-free. Therefore, the final version, after possible acceptance, will be corrected by the proofreading service in the MDPI.

We look forward to hearing from you in due time regarding our submission and to responding to any further questions and comments you may have.

Best Regards

Authors

Reviewer 3 Report

The article discusses some problems with the security of the access control system, more precisely Radio Frequency Identification realized on Low Frequency. The authors analyze the possibilities of intercepting the electromagnetic message from the RFID device and its reproduction in order to ensure unauthorized access to workplaces, residential buildings, time gate control, etc.

The presentation of the article is distinguished by a clear and understandable style. The article is richly illustrated with well-crafted images and graphics that complement the presentation and make it easier to understand. The obtained results are convincingly presented.

The article will definitely generate interest in the RFID audience due to its informative nature.

As the main shortcoming of the article, I can point out the absence of significant scientific contributions. Almost all conclusions are trivial from engineering practice. This is also the main reason for the low final estimate of the article.

The article does not need substantial editorial corrections. However, there are some spelling mistakes that should be corrected.

Author Response

Dear Reviewer

Thanks for providing the detailed and constructive review for our paper which allowed to increase a scientific level of this paper. We tried to reply for each point mentioned in the review. Our corrections were highlighted in manuscript by blue and red colours. We hope that the new version of our manuscript looks better and meets your requirements.

Here we’d like to reply for each point mentioned in the review:

 As the main shortcoming of the article, I can point out the absence of significant scientific contributions. Almost all conclusions are trivial from engineering practice. This is also the main reason for the low final estimate of the article.

Our response

Engineering measurements are used for data acquisition, which are used to analyze the electromagnetic safety of short-range radio frequency identification systems and conclusions are drawn on the base of them. Such a way of proceeding is characteristic of the way of conducting empirical research, which in our opinion is more valuable than the methods of conducting analytical research or mathematical or computer modeling, because they have a strong direct reference to reality.

The added value, in our opinion, is the way to improve the signal quality and extend the range of communication using the resonance phenomenon, as well as the use of band-pass filtration aimed at cutting out the components of the spectrum carrying (carrier) information and amplifying the information components.

We have revised the manuscript from the abstract to the conclusions. We add new abstract and defined more precisely motivation and goals.

We also tried to draw more sophisticated conclusions. We hope that it influence on increasing your evaluation.

In addition to the above comments, we conducted extensive editing of English language and style. We realize that, despite all our efforts, this article may not be language and style error-free. Therefore, the final version, after possible acceptance, will be corrected by the proofreading service in the MDPI.

We look forward to hearing from you in due time regarding our submission and to responding to any further questions and comments you may have.

Best Regards

Authors

Round 2

Reviewer 3 Report

Please check Fig.3 RFID LF card reader with one transmitting coil and one receiving coil.

It is incomplete. A picture absent!